# Cockle Population Dynamics in a Complex Ecological Aquatic System

**DOI:** 10.3390/biology14101427

**Published:** 2025-10-17

**Authors:** Simão Correia, Marta Lobão Lopes, Ana Picado, João M. Dias, Nuno Vaz, Rosa Freitas, Luísa Magalhães

**Affiliations:** 1CESAM (Centre for Environmental and Marine Studies), Departamento de Biologia, Campus Universitario de Santiago, 3810-193 Aveiro, Portugal; 2CESAM (Centre for Environmental and Marine Studies), Departamento de Física, Campus Universitario de Santiago, 3810-193 Aveiro, Portugal

**Keywords:** Cardiidae, *Cerastoderma edule*, Ria de Aveiro, recruitment, density, growth

## Abstract

**Simple Summary:**

The European cockle is a key shellfish in Portugal’s Ria de Aveiro, supporting both nature and local livelihoods. This study looked at where and when cockles are most common, how they grow, and where young cockles settle. We found differences across the lagoon: in some areas cockles were absent, while in others they were extremely abundant, especially in summer and autumn. Their numbers and growth were strongly shaped by water salinity, temperature, currents, and sediment. Alarmingly, the average size of cockles was close to the legal minimum for harvest, raising concerns about overfishing. These results show that both environmental conditions and harvesting pressure put the cockle population at risk. To secure this resource for the future, better management is needed, including strict compliance with minimum catch size, seasonal harvest bans, and habitat protection.

**Abstract:**

*Cerastoderma edule*, the European edible cockle, is a key species in the coastal ecosystems of Portugal, particularly in Ria de Aveiro, a biodiversity hotspot and a critical area for cockle harvesting. This study aimed to assess the population dynamics of *C. edule* in Ria de Aveiro, focusing on spatial and seasonal patterns in density, growth, cohort composition, and recruitment areas, to provide baseline data for sustainable management. Our results revealed marked spatial and seasonal variability in cockle density, ranging from complete absence at some upstream sites to peaks of over 5900 ind. m^−2^, with recruitment concentrated in summer and early autumn. Environmental gradients, particularly decreasing salinity inland, seasonal temperature shifts, and current velocity, strongly shaped the distribution of recruits and adults, while cohort lifespan and growth performance varied with sediment conditions and lagoon position. Concerningly, the maximum mean shell length observed is close to the legal minimum catch size, raising questions about population sustainability under current harvesting pressures. This interplay of environmental drivers and harvesting pressures poses risks to population viability. Effective management strategies, including adjusted catch sizes, seasonal harvesting bans, and habitat conservation, are essential to ensure the sustainable exploitation of cockles in Ria de Aveiro. Enhanced research and monitoring efforts are recommended to support informed management decisions and protect this valuable resource.

## 1. Introduction

*Cerastoderma edule*, the European edible cockle, is distributed along the Atlantic coast of Europe and northwest Africa [1], being typically found in tidal flat coastal systems such as bays and estuaries. Cockles are suspension feeders, consuming phytoplankton, zooplankton, organic debris, and even conspecific larvae, with dietary preferences shifting from small phytoplankton in larvae to organic-rich material in adults [2]. Reproduction is gonochoric with external fertilisation. Gametogenesis begins in late winter, spawning occurs from May to August, and maturation typically happens at around 18 months and 15–20 mm shell height [3]. Growth and survival are shaped by both biotic factors like food, predation, and parasitism, and abiotic ones such as temperature, hydrodynamics, and sediment conditions, all of which interact in complex ways without a single dominant driver [4].

Cockles are extensively exploited by some European countries, namely Portugal and Spain, with total European captures averaging 20,000 tonnes per year during 2013–2023 [5], generating an estimated revenue of more than 40 M€. Despite the reduced continental area, Portugal represents the third largest producer of cockles in Europe, with more than 13% of total live weight captures [5]. Cockle catches declined steadily from the 1950s to the early 1990s, then rebounded strongly, reaching record highs from 2003 to 2023 [5]. In the most recent decade, Portugal captured an average 2681 tonnes of cockles per year [5]. Translated into income, cockles represented more than 6 M€ out of 285 M€ retrieved from all species and 91 M€ from molluscs in 2020 [6].

In Portugal, the coastal lagoons that mostly contribute to such catch and income numbers are Ria de Aveiro, followed by Ria Formosa, with the first supplying more than half of the cockle tonnes captured each year [6]. Indeed, it is no coincidence that Ria de Aveiro is considered one of the most important European hotspots of biodiversity [7]. High densities of *C. edule* supports high biodiversity [8] due to their importance for example, to trophic webs [9,10], ecosystem services (e.g., [11]), biogeochemistry [12], and biogeomorphology [13]. In the Ria de Aveiro, cockles (*Cerastoderma edule*) are preyed upon by crabs, fish (e.g., plaice, eel, seabass) and birds (e.g., seagulls, cormorants, flamingos) [2], while fatty-acid studies confirm their role as a key trophic link transferring benthic primary production to higher consumers [11,14,15]. Moreover, cockle bioturbation has been shown to reduce microphytobenthic biomass through sediment disturbance, although this effect varies with sediment type due to a balance between biofilm disruption and nutrient stimulation from bioirrigation [16].

The exploitation of this aquatic system in environmental, economic, social, cultural, and recreational terms is of strategic importance at a regional and national level [17,18,19]. Furthermore, cockle harvesting is often highlighted as the most important traditional activity performed in this coastal lagoon, with a proportional strong cultural footprint [2].

The rise of serious European transversal threats such as illegal, unreported, and unregulated fisheries [20], alongside the emergence of diseases [21] such as Marteiliosis [22] or trematode parasites [23], has been contributing to the decline of natural stocks of the European edible cockle. Given its importance for the European economy, and the acknowledgement of its crucial ecological role [11], this decline has spurred consistent scientific interest in this species. For the last 10 years (2013–2023), publications with “*Cerastoderma edule*” as the main topic suffered a 2-fold increase (Data derived from Clarivate InCites. © Copyright Clarivate 2023. All rights reserved.). Recent studies cover a wide range of subjects, from cockle traceability, hatchery production, disease dynamics, and even whole genome description. More in detail, cockles were used as a biological model to develop traceability innovative tools [24,25,26], an essential instrument for tracing, tracking, and reducing the mislabelling of food items throughout the entire supply chain. In line, and due to a growing market demand, cockles hatching methodologies have received increasing attention and improvements [27]. Disease inventory, dynamics recognition, and possible mitigation measures are other hot topics [8,28,29]. Consequently, the expansion of cockle-related state of the art has contributed to the development of better and more efficient common and participatory management strategies, crucial for this important biological resource. Nevertheless, the data available for each European country is highly unbalanced, primarily related to different investment capacities in R&D actions, including human resources. In Ria de Aveiro, despite some dispersed and/or opportunistic studies with quite important insights about cockle population dynamics and distribution [3,30], local ecological knowledge [2,31], host–parasite dynamics [8,32], and bioaccumulation patterns [33], consistent long-term monitoring of cockle population is lacking. For example, Maia et al. [3] conducted a two-year study (2013–14) reporting on growth, reproduction, size at first maturity, and condition index, while Matos et al. [30] modelled habitat suitability under current and projected environmental scenarios, showing how salinity, submersion time, and freshwater discharge shape cockle distribution. In contrast, consistent long-term monitoring entails standardised surveys conducted over extended periods (often exceeding a decade), allowing detection of interannual fluctuations, long-term trends, and responses to climatic or anthropogenic pressures, which is currently lacking for the studied area.

Thus, to overcome this gap, the present study aims to provide comprehensive, long-term information on the population dynamics of *Cerastoderma edule* in Ria de Aveiro. Specifically, we investigate cockle density, cohort composition, and growth patterns, as well as identify key recruitment areas and stock beds, in order to inform sustainable management and conservation strategies for this ecologically and economically important species.

## 2. Methods

### 2.1. Study Area and Sampling Strategy

The study area, Ria de Aveiro, is a coastal lagoon with a maximum width and length of 10 and 45 km, respectively, located on the northwest coast of mainland Portugal, between 40°38′ N and 40°57′ N (Figure 1). The connection between the lagoon and the Atlantic Ocean is made throughout an artificial inlet, resulting in semidiurnal tides with approx. 2 m range at the inlet [34]. This mesotidal coastal lagoon is a shallow, temperate, and well-mixed system [18], with an area between 66 and 82 km^2^ depending on the tide [35]. Ria de Aveiro is formed by four main channels, São Jacinto-Ovar, Espinheiro, Ílhavo, and Mira, organised into several branches that form inner basins, mudflats, and small islands [36]. Due to the combined effect of freshwater discharges and tidal penetration, Ria de Aveiro exhibits a longitudinal gradient of salinity and temperature. Despite all channels receiving freshwater inputs, the major fluvial inputs come from Vouga and Antuã rivers [35]. Cockle banks are typically co-inhabited by polychaetes (e.g., *Tharyx* spp.), other bivalves (e.g., *Scrobicularia plana*), gastropods (e.g., *Peringia ulvae*), and arthropods (e.g., *Cyathura carinata*), whose diversity and abundance tend to increase with cockle abundance, although this relationship is clearly influenced by the underlying abiotic conditions [8].

To achieve good coverage of the cockle’s distributional range, specimens were collected monthly in intertidal zones at low tide throughout one year, from June 2020 to May 2021, at 18 intertidal sampling locations spread across the four main channels of the lagoon (18 sampling locations × 12 months). Sampling locations were named using channel name first letter (S for São Jacinto-Ovar, E for Espinheiro, I for Ílhavo, and M for Mira) followed by L to designate “sampling Location” and a number from 1 to 18 (Figure 1). Location 7 (SL7) was only sampled in 6 out of the 12 sampling times due to tidal submersion.

Cockles were collected using the classic quadrate method, which is widely used for assessing bivalve population dynamics and offers reliability comparable to more modern techniques [37]. The first 5 cm layer of sediment was removed from six 0.25 m^2^ PVC quadrates aligned in a 100 m transect parallel to the waterline. The sediment of each quadrate was washed in situ through a 1 mm mesh sieve, placed separately in a plastic container, and transported to the laboratory.

In the laboratory, cockles were hand-sorted and counted, and their shell length was measured with a digital calliper (IP67 MITUTOYO^®^ Kawasaki-shi, Japan) connected to a computer. Measurements were rounded to the last millimetre and assigned to a shell length class. Abundance (total number of individuals) per sampling location and sampling time (18 locations × 12 months) was transformed into density (individuals per m^2^) based on the sampled area per location (6 quadrates × 0.25 m^2^).

### 2.2. Environmental Characterisation

Several abiotic estuarine variables were obtained using the hydrodynamic and water quality numerical model DELFT3 D, considering the FLOW and WAQ modules. The model was implemented and validated for Ria de Aveiro [38]. In this study, model outputs were compared with in situ measurements of water level, currents, salinity, temperature, dissolved oxygen, pH, chlorophyll-a, and nutrient concentrations, showing good agreement and thus confirming the model’s reliability in simulating the hydrodynamic and biogeochemical conditions of the lagoon. The model provides information for the hydrodynamical setting and water biogeochemical variables in an irregular horizontal grid with a spatial resolution ranging from 50 to 100 m. A one-year simulation was carried out, covering the period from June 2020 to May 2021, proceeded by a six-month spin-up period to ensure model stabilisation. Monthly mean values of each environmental variable, chosen for their relevance on cockles’ distribution [30], were extracted at the 18 sampling locations. The computed variables included mean and maximum salinity, mean and maximum temperature, mean and maximum current velocity, mean chlorophyll-a, and total submersion time.

In each sampling location and time point (18 × 12), sediment samples were collected in triplicate for grain size and total organic matter (TOM) analyses. Grain size analysis was carried out by wet sieving of the fines content (i.e., sediment with particles below 0.063 mm after treatment with increasing concentrations of hydrogen peroxide for organic matter removal) and dry sieving of the remaining fractions (0.063–0.125 mm, 0.125–0.250 mm, 0.250–0.500 mm, 0.500–1 mm, 1–2 mm) [39]. The grain-size fractions were expressed as a percentage of the whole sediment and the data was used to calculate the median particle diameter value, P50, in phi (Φ) units. Sediments were classified based on the median grain size (MGS) according to the Wentworth scale: very fine sand (median from 3 to 4 Φ); fine sand (2–3 Φ); medium sand (1–2 Φ); coarse sand (0–1 Φ); very coarse sand (−1–0 Φ) [40,41]. For the TOM analysis, sediment was dried at 60 °C for 24 h and grounded to powder using a mortar and a pestle. After combustion of 1 g of dry sediment at 450 °C for 4 h, the TOM content was determined as the percentage of weight loss on ignition [42]. TOM content was considered very low when below 1%, low between 1 and 2%, and medium between 2 and 4%, a widely used classification for assessing organic enrichment in coastal systems (e.g., [43]).

### 2.3. Data Analysis

#### 2.3.1. Cockle Density Distribution According to Environmental Variables

Differences in density between sampling locations and time were tested using a two-way ANOVA (18 location × 12 months) in the R statistical programming language (version 4.3.1), after applying a square root transformation and confirming homogeneity of variance with a Levene’s test. Location and time were included as main effects, with an interaction term indicating whether the magnitude of change throughout the year differed among sampling locations.

Generalised Linear Mixed Models were fitted using the lme4 package [44] in the R statistical programming language (version 4.3.1) to evaluate the relationship between cockle density (total, recruit, and adult stages) and a suite of environmental predictors. Models were constructed separately for each life stage using log-transformed density as the response variable to normalise residuals and reduce skewness. For the total density model, salinity and temperature were included as fixed effects. The recruit density model incorporated salinity, temperature, MGS, water current velocity, and month as fixed effects, while the adult density model included salinity, temperature, and TOM. Sampling location was included as a random intercept. Only variables that were statistically significant were retained in the final models. Model assumptions were evaluated by inspecting residual plots and assessing normality and homoscedasticity.

Monthly spatial means over the entire model domain were computed and mapped using MATLAB (version R23b), applying standard toolboxes. Only statistically significant model outputs were mapped. MGS and TOM, while significant predictors in the models, were not mapped due to their site-specific and non-interpolated nature. Recruit and adult densities were represented spatially (at the 18 locations) through proportional circles, with bubble sizes scaled relative to a fixed reference maximum of 3000 ind.m^−2^. The two life stages were distinguished using different colours.

#### 2.3.2. Descriptive Analysis of Cockle Population Dynamics

This population analysis focused on the following parameters: age composition, recruitment, growth parameters, total mortality rate, stock size (number of individuals), and biomass (g DW m^−2^). Length–frequency histograms were performed and analysed using the TropFishR package [45] in the R statistical programming language (version 4.3.1). Bhattacharya’s method was used to derive age groups from length–frequency histograms. This method uses modal progression analysis to identify individual-size cohorts as individual normal distributions within a composite distribution of multiple age groups [46]. Cohorts were assumed to be single when the separation index was >2, commonly used to indicate that cohorts are considered distinct [47]. For each site, the Bhattacharya analysis provided the mean length, standard deviation, and relative abundance of each cohort. Cohorts were then assigned to life stages based on their size ranges, guided by previous studies and field observations: recruits (newly settled individuals), juveniles (first-year cockles), and adults (second-year or older cockles). Data are not available for location L18 because the method could not be applied accurately with such low densities.

ELEFAN method was used to derive growth parameters of the von Bertalanffy growth function (VBGF) from length–frequency data [48]. For that, length–frequency data was restructured by scoring length bins based on deviations from a moving average across neighbouring bins. Then, the cumulative score for a given set of VBGF parameters based on the bin scores that are intersected was calculated, resulting into growth curves. To finalise, VBGF parameters that resulted in the maximum score value were assessed, while confidence intervals were calculated with the jack knife technique with replacement [49]. A VBGF was obtained for each sampling location as well as a growth performance index (Lt) [50]:

Lt = L∞ [1 − e − K(t)], where L∞ is the asymptotic shell length (mm) and K is the growth coefficient (year^−1^).

Using the R (version 4.3.1) package TropFishR as well, recruitment patterns were assessed from length–frequency data and withdraw in number and percentage. Length-converted linearised catch curves were applied to age composition and length–frequency data to estimate the total mortality rate (Z). The overall stock size in numbers and biomass (gDW m^−2^) was also calculated using the virtual population analysis function [51].

A correlation matrix was obtained for VBGF parameters, cockles’ density (total, mean, and recruitment peak), and population dynamics calculated variables (Z, stock size and biomass).

## 3. Results

### 3.1. Environmental Characterisation

Mean values of environmental predictors computed are available in Table 1. Salinity showed a clear decreasing gradient from the outer lagoon (SL1: 31.5 ± 2.8) toward the southernmost Mira channel (ML18: 20.0 ± 12.7). Temperature ranged moderately across sampling locations (16.0 ± 2.5 °C to 18.2 ± 4.9 °C), with slightly higher values observed inland (e.g., SL9 and ML18). Current velocity varied markedly, with the highest values observed at constricted or channel-narrowing sites (EL11: 0.97 ± 0.01 m/s; SL9: 0.86 ± 0.01 m/s), while more sheltered areas such as ML15 and EL6 exhibited the lowest velocities (0.09 ± 0.00 and 0.16 ± 0.00 m/s, respectively). Chlorophyll-a concentrations were highly spatially variable, with the highest values observed at the southern Mira channel (ML18: 5.3 ± 4.8 µg/L), followed by southern-central sites (ML17: 3.9 ± 3.5 µg/L; ML16: 3.0 ± 2.6 µg/L). Concentrations were lower at central lagoon locations (e.g., EL6: 2.1 ± 1.7 µg/L; EL10: 1.9 ± 1.5 µg/L), and generally below 2.0 µg/L at most other sites. Total submersion time was largely stable (730.0 ± 21.6 h) across most locations, except at SL3 (469.6 ± 13.4 h) and ML15 (557.2 ± 15.9 h).

Table 2 presents the spatial variation in sediment median grain size (MGS) and organic matter content (TOM) across the 18 sampling locations. Most sampling locations were characterised by fine sand, particularly in the outer and central lagoon areas (SL1, SL2, SL4–EL6, SL9–IL12, IL14). Very fine sand was observed exclusively at station EL11 in the Espinheiro channel. In contrast, medium sand was prevalent in southern and some mid-lagoon stations, notably SL3, SL7, SL8, IL13, and within the Mira channel (ML15, ML16, ML18). TOM content ranged from 0.9% to 3.4%. Medium TOM levels (2–4%) were primarily observed in mid-lagoon and southern zones, particularly at stations SL1, SL4–SL6 (São Jacinto/Ovar), EL11 (Espinheiro), IL14 (Ílhavo), and ML17 (Mira). The highest TOM values were recorded at SL5 and EL11, both measuring 3.4%, suggesting organic matter accumulation in low-energy, depositional environments. Low TOM levels (1–2%) were dominant in inland and southern areas including stations SL3, SL7–EL10, IL12–IL13, ML15, and ML18. Very low TOM content (<1%) was observed only at station ML16 (0.9%), the most inland site within the Mira channel.

### 3.2. Population Dynamics

#### 3.2.1. Cockle Density Distribution According to Environmental Variables

The maximum total density was 5932 ind./m^2^ for location ML17 in September 2020. Excluding the total absence of cockles (0 ind./m^2^) registered at locations SL9 (from December 2020 to April 2021) and ML18 (in July 2020 and from January to March 2021), the minimum total density was 1 ind.m^−2^ for location SL9 (in July 2020 and from September to November 2020) and ML18 (in August, October and December 2020 and April 2021) (Figure 2). Locations nearest the entrance of the lagoon (SL1, SL2, SL3, SL4, IL12 and ML15) presented high cockle mean density (>800 ind.m^−2^). Low cockles mean density (<400 ind.m^−2^) was registered in the upstream locations (SL8, SL9 and ML18). The remaining sampling locations (SL5, EL6, SL7, EL10, EL11, IL13, IL14, ML16 and ML17) showed an intermediate mean density (400–800 ind.m^−2^). Overall, the mean density was higher (>800 ind.m^−2^) in June and between August and October 2020. The lowest mean density was registered between January and April 2021 (<400 ind.m^−2^). All remaining months displayed intermediate values of mean cockle density (400–800 ind.m^−2^). The seasonal trend was not verified for sampling locations SL7, SL8, SL9, and ML18, mainly due to low total density (Figure 2). Both factors, sampling location and month, presented a significant influence on density with a significant interaction (two-way ANOVA: F_(17)_ = 11.0, *p* < 0.001, F_(11)_ = 20.0, *p* < 0.001, F_(183)_ = 2.1, *p* < 0.001, respectively).

Generalised Linear Mixed Models revealed distinct relationships between environmental variables and cockle density across life stages: total, recruit, and adult (Table 3). Both salinity and temperature were significant positive predictors of total cockle density. Salinity had a stronger effect (Estimate = 0.068, *p* < 0.001) than temperature (Estimate = 0.072, *p* = 0.013). This suggests that areas with higher salinity and temperature supported higher overall cockle densities. Recruit density was significantly influenced by multiple environmental factors. Temperature (Estimate = 0.280, *p* < 0.001) was a strong positive predictor, as well as salinity but with a lower significance (Estimate = 0.070, *p* = 0.032). Current velocity (Estimate = −2.384, *p* = 0.063) and MGS (Estimate = 0.742, *p* = 0.073) had a marginally negative and positive effect, respectively, suggesting a weaker influence. Month was also included as a categorical variable and showed an overall significant effect, indicating temporal variation in recruit density. For adult cockles, salinity (Estimate = 0.061, *p* = 0.002), temperature (Estimate = −0.102, *p* = 0.002), and TOM (Estimate = 0.343, *p* = 0.003) were all significant predictors. Interestingly, temperature exhibited a negative relationship with adult density, contrasting with its positive effects on total and recruit densities.

For their significant influence on cockles’ density, salinity, temperature and current velocity were mapped. Salinity (Figure 3) exhibited a pronounced spatial gradient throughout the year, consistently decreasing from the northern marine-influenced areas toward the southern and inner lagoon zones. Seasonal variation was also evident: highest salinity levels (>30) occurred in the northern region during summer (June–August 2020) and again in late spring (May 2021), while lower salinity values (<15) were recorded in the southern areas during winter and early spring (December–March). Cockle recruits were primarily found from June to October 2020, particularly in zones with intermediate salinity (15–25), suggesting a preference or tolerance for mesohaline conditions during settlement. Adult distributions appeared broader and more stable, occurring across a wider range of salinities, though with higher densities generally in regions with moderate to high salinity.

Seasonal temperature fluctuations were clearly delineated (Figure 4), with summer months (June–September 2020) exhibiting the highest temperatures (>22 °C), particularly in the northern and mid-lagoon regions. A sharp decline in temperature began in October, reaching annual minima during winter (December 2020–February 2021) with values dropping below 13 °C in the southern and inner lagoon zones. Temperatures then gradually increased from March to May 2021. Cockle recruits were predominantly present during warmer months (June to October 2020), with peak densities observed in August and September when water temperatures ranged from ~19 °C to 23 °C. These recruits were mainly located in the central and southern portions of the lagoon, coinciding with moderately warm areas. Adult cockles were found throughout the year, but densities were highest during cooler months (October–February), particularly in the southern and central regions.

Velocity patterns (Figure 5) exhibit clear seasonal variation, with higher intensities generally observed in late spring and summer months (e.g., June–August 2020, and May 2021), particularly along the central channel and southern sections of the lagoon. Lower velocities prevailed during winter months (December–February), especially in the northern and inner lagoon zones. It is possible to observe some overlap between high current velocities and reduced recruit densities. Adult distribution appeared more stable and less associated with short-term velocity shifts.

#### 3.2.2. Cohort Analysis

Cockle population of Ria de Aveiro coastal lagoon often consisted of 3 cohorts (8 out of 17 sampling locations, excluding ML18 from the analysis): recruits (1–9 mm, the newly settled cockles), juveniles (9–17 mm, cockles in their first year), and adults (17–26 mm, second year cockles). A fourth cohort was registered at 6 sampling locations, while other 2 locations presented only 1 cohort. Sites with a greater number of year-class cohorts tended to support higher total and mean cockle densities (R = 0.51, *p* ≤ 0.05), suggesting that population structure and recruitment variability contribute significantly to local abundance.

Sampling locations SL2 and IL12 where the ones presenting the maximum mean shell length, 26.4 ± 1.6 mm and 26.2 ± 2.6 mm, respectively. These sampling locations also had the longest cohort lifespan (Figure 6): 11 and 12 months, respectively. In most sampling locations, the cohort lifespan was shorter, i.e., 4 to 9 months (average = 4.5 ± 3.0 (sd)). Cohort maximum lifespan was significantly and positively correlated with cockle total and mean density (R = 0.62 and R = 0.59, *p* ≤ 0.05), and median grain size (R = 0.47, *p* ≤ 0.05). Temperature was significantly negatively correlated with cohort maximum lifespan (R = −0.50, *p* ≤ 0.05). There was also a moderate but still significant negative correlation between cohort maximum lifespan and the recruitment month (R = −0.51, *p* ≤ 0.05).

#### 3.2.3. Growth

Growth parameters L_∞_ and K, calculated for each sampling location, ranged from 24.3 to 47.9 mm and from 0.5 to 0.7 year^−1^, respectively (Table 4). The growth performance index fluctuated between 2.5 and 3.1 (Table 4). The latter and L∞ were significantly and negatively correlated with the number of cohorts (R = −0.52 and R = -0.53, respectively, *p* ≤ 0.05), and with the position of the sampling locations along the coastal lagoon (R = −0.48 and R = −0.51, respectively, *p* ≤ 0.05), with higher growth rates observed in the northern branches.

#### 3.2.4. Recruitment, Mortality, Stock Size and Biomass

In density, recruitment peaks ranged from 542 (L6) to 5932 (L17) ind.m^−2^ (37 and 8 ind.m^−2^ registered for SL9 and ML18, respectively were not considered for the range). Most recruitment peaks occurred in June (31.3%) and September (31.3%), followed by August (25%), while a single recruitment occurrence was registered in October (SL5) and November (IL13). The strongest recruitments (>2000 recruits.m^−2^) occurred south from the Ria’s mouth (in the Mira channel), while lower recruitment peaks (542–985 recruits.m^−2^) were mainly registered in the Espinheiro channel, located northeast of the Ria’s Ocean boundary. The remaining sampling locations presented intermediate recruitment densities that ranged from 1000 to 1800 ind.m^−2^. A clear relationship between recruitment month and recruitment density was not evident despite a higher mean recruitment density (2328.0) occurring in September comparing to June and August (1428.6 and 1160.3, respectively). Recruitment peak was significantly and negatively correlated to cohort maximum lifespan (R = −0.51, *p* ≤ 0.05) and dependent on the sampling location (Lat: R = −0.69, *p* < 0.01, Lon: R = −0.71, *p* < 0.01).

Mortality (Z) ranged from 0.9 to 1.8 for sampling locations ML15 and SL4, respectively (Table 5). Minimum stock size and biomass were calculated for sampling location ML17 (0.59 and 0.85 g DW m^−2^, respectively) while maximum values were achieved for location SL4 (288.12 and 106.63 g DW m^−2^, respectively) (Table 5). Mortality was significantly and positively correlated with both stock size (R = 0.72, *p* < 0.01) and biomass (R = 0.80, *p* < 0.001). Stock size and biomass were significantly and positively correlated to TOM (R = 0.55 and R = 0.54, respectively, *p* ≤ 0.05).

## 4. Discussion

In Ria de Aveiro, the distribution and density of *Cerastoderma edule* are strongly shaped by environmental conditions. Our findings identify water salinity and temperature as the primary drivers of cockle density in this aquatic system, with sediment characteristics and current velocity also playing a role, although to a lesser extent. This multifactorial influence aligns with existing literature, which recognises that bivalve distribution along coasts and estuaries is shaped by a combination of environmental gradients (e.g., [52]).

Salinity stands out as a physiologically limiting factor for estuarine species [53,54]. For bivalves, salinity outside the species-tolerance range can supress activity and feeding, while simultaneously increasing the energetic cost of osmoregulation [55,56]. *C. edule* tolerates a wide range of salinities, from 12.5 to 38.5 [57], with salinities above 22 offering progressively more suitable conditions [30]. Larval stages are more sensitive, while survival is possible at salinities of 40, metamorphosis fails at 45, and deformities may occur at salinities of 20–25 [57]. In an estuarine environment like Ria de Aveiro, organisms are naturally exposed to daily (short-term) and seasonal (long-term) fluctuations in water salinity, due to tidal flows and freshwater inputs, both intrinsically related to the system morphology and hydrodynamics [58]. Our monthly sampling design captures seasonal variability, though short-term (e.g., tidal) fluctuations remain unresolved, which may limit interpretation of temporal effects. In Ria de Aveiro, a pronounced horizontal salinity gradient defines the transition from marine-influenced areas to the upper fluvial zones, where freshwater dominates despite semidiurnal tides [59]. Seasonally, salinity rises in the dry summer months and drops during wetter periods [60], expected to influence cockle distribution accordingly. Indeed, densities were higher near the sea, while inner zones showed sparse populations. From January to April 2021, cockle densities declined in response to winter freshwater influx. The spatial overlap between cockle recruits and intermediate salinity zones, especially in late summer and early autumn, supports the hypothesis that salinity gradients may influence early life stage distribution and survival. This pattern aligns with established knowledge on salinity-mediated recruitment and displacement in estuarine bivalves such as *Venerupis corrugata* (reviewed in [61]) and other infaunal bivalves [62]. Conversely, adult cockles appeared less sensitive to salinity fluctuations, indicating higher tolerance, as suggested by Peteiro et al. [56] or greater site fidelity. This trend reflects what is described for Portuguese [30,63] and other European estuaries [57,64,65]. Abrupt salinity shifts, often triggered by heavy rainfall, have been linked to cockle mass mortality events [66,67,68]. With climate change expected to bring increased rainfall and warming [69,70], Ria de Aveiro’s upstream cockle populations may face heightened risk from low salinity stress [71].

Although cockles tolerate temperatures between 4 and 38 °C [72], tolerance is influenced by season and physiological condition. In Ria de Aveiro, habitat suitability models suggest optimal temperatures below 23 °C, ranging from 16 to 17.5 °C in spring and from 20 to 23 °C in summer [30]. Similar patterns are reported for the Mondego estuary (Portugal), where survival declines above 28 °C [54]. Our results reflect this thermal sensitivity, since adult cockle densities decline with higher temperatures, likely due to increased metabolic stress. For example, as seen in Zhou et al. [73] where long-term heat exposure led to elevated respiration rates, reduced health condition, and ultimately mortality, suggesting that cockles under thermal stress allocate more energy to basic maintenance and less to growth and survival. Conversely, recruit densities rise with temperature, suggesting favourable conditions for larval development and settlement during warmer periods as was already demonstrated in hatchery conditions [74]. These dynamics reflect seasonal recruitment pulses coinciding with warmer months. Together, these patterns indicate a strong seasonal signal in temperature that likely governs the timing of recruitment events and influences juvenile distribution, while adult cockles persist across a wider thermal range. This suggests that while recruitment appears to be closely linked to warmer temperatures, likely related to spawning and settlement timing [75], adults exhibit broader thermal tolerance and occupy more stable habitats year-round [76]. Temperature and salinity are closely linked in estuarine systems and are modulated by tidal regimes [77]. In Ria de Aveiro, oceanic-influenced habitats, characterized by higher salinity, also tend to experience more thermally buffered conditions. This environmental dynamic helps explain the spatial and seasonal variation in cockle density. These findings are consistent with similar patterns observed in other estuarine ecosystems [78,79]. It is important to note that, while thermal buffering mitigates extremes, regional differences in absolute temperatures still play a crucial role in population structure.

The identification of up to four distinct cockle cohorts across sampling sites suggests that population complexity may reflect not just reproductive success but also habitat stability and suitability. In estuarine systems, where environmental variability can impose episodic recruitment failure, the presence of multiple cohorts may indicate more consistent and favourable conditions. The spatial pattern observed, with locations such as SL2 and IL12 supporting larger individuals and extended cohort lifespans, may be attributed to site-specific environmental conditions, likely more stable salinity, that allow cohorts to persist longer. Conversely, the observed negative correlation between temperature and cohort lifespan points to the potential stress imposed by elevated thermal conditions, which may reduce survival or increase turnover rates, especially in warmer months. This aligns with known effects of temperature on bivalve metabolic rates and life expectancy [76] and supports the hypothesis that rising temperatures under climate change may alter population structure by favouring shorter-lived, faster-growing cohorts.

Growth parameters (L∞ and performance index) were negatively correlated with cohort number and latitude, which reflect the spatial variation in food availability, hydrodynamics, or other abiotic drivers across the lagoon. The lower L∞ values in areas with more frequent cohort replacement suggest a trade-off between growth potential and recruitment success. When compared with other regions [64], our estimated values are lower than those reported for the Wadden Sea (ɸ’ = ~3.0) and comparable to populations from northwest Spain (ɸ’ = ~3.05) and the British Isles (ɸ’ = ~2.7–2.8). Higher growth performance values have been observed in a French Atlantic Bay (ɸ’ = ~3.3 [47]), reflecting faster growth and larger minimum sizes under favourable conditions. The lower growth performance observed in many of our lagoon sites likely result from trade-offs between frequent recruitment and reduced individual growth potential, as also suggested in other studies of *C. edule*. Overall, these comparisons highlight that spatial variation in cockle growth across our lagoon is consistent with broader biogeographic patterns, where local differences in food availability, hydrodynamics, temperature, and population density strongly influence growth trajectories. Such heterogeneity underscores the importance of accounting for life-history traits in management strategies, since areas supporting fast-growing individuals (e.g., IL14) may require different harvesting thresholds to maintain sustainable yields compared with sites dominated by smaller, slower-growing cockles.

Sediment type plays an important role in shaping the habitats suitable for cockles, particularly by influencing settlement, burrowing, and feeding efficiency. Cockles live buried just at the interface between the sediment and the water column, inhabiting clean sand, muddy sand, mud, or muddy gravel bottoms, with a general preference for muddy sand to sandy-mud sediments [80]. In our study, recruit densities were positively associated with finer sediments (i.e., lower median grain size), suggesting that early life stages benefit from the stability and food availability of muddier habitats. These sediments may enhance settlement success and reduce the risk of dislodgement, offering favourable microhabitats for juvenile development [81,82]. Significant densities of cockle settlement are found in the Bay of Somme (France) when granulometric characteristics are between 50 and 200 µm [83], while in the Bay of Saint-Brieuc (France), it ranges between 100 and 125 µm. In the Wadden Sea (Netherlands), optimal recruitment success is observed at mud contents of about 0.5 to 3% in late winter, which corresponds to 1 to 10% in summer [84]. The silt content of the sediment will certainly improve water retention, which is crucial for the survival of young cockles [83]. Interestingly, adult densities were positively related to organic matter content, despite initial expectations that higher organic load reduce sediment oxygenation. However, organic matter levels in our study were relatively low (0.21% to 5.30%), likely below thresholds that would impair oxygen diffusion. Instead, moderate organic enrichment may improve food availability by supporting detritus and microbial productivity, important dietary components for adult cockles [85]. This indicate that we studied a particular type of habitat characterized by low organic matter input but suggests that even subtle differences in organic content can influence adult distribution. Overall, sediment characteristics appear to influence different life stages differently, with finer particles benefiting recruits and moderate organic matter favouring adults. These findings have significant implications in the context of cockle population management in Ria de Aveiro. Changes in sediment composition and distribution, which are expected to occur in climate change scenarios, and other anthropogenic-derived pressures such as dredging activity to maintain navigable depths in the harbour [86], can disrupt the delicate balance of mud content necessary for optimal recruitment success of cockles. This could lead to fluctuations in population sizes and potentially impact the broader marine food web and coastal fisheries.

Current velocity had no clear effect on adult density, but recruit densities showed a weak negative correlation, implying that stronger currents may hinder settlement or cause post-settlement dislodgement. Flume experiments support this hypothesis, demonstrating current-mediated impacts on larval adhesion and recruitment [87,88]. Hydrodynamic changes also shape sediment distribution and stability, meaning that any shifts in current regimes driven by climate change or human interventions could have cascading effects on habitat suitability and recruitment.

Cockle recruitment and cohort persistence depend on a complex interplay of environmental and biological processes. In Ria de Aveiro, August seems to be the most favourable month for recruitment, likely due to hydrodynamic factors. During this period, the region experiences stronger winds and increased tidal activity, which enhance water movement and nutrient availability [34]. This higher hydrodynamic activity is crucial for the dispersal and settlement of larvae [4]. The summer months, particularly August, coincide with the peak of the Atlantic’s thermal stratification. This brings nutrient-rich deeper waters to the surface, fostering a more productive environment for larvae. Moreover, the wind patterns in August, characterized by stronger northerly and north-westerly winds, contribute to coastal upwelling. This phenomenon further enhances nutrient availability and creates favourable conditions for recruitment, as larvae benefit from both increased food supply and suitable settlement habitats. However, high recruitment does not necessarily guarantee a strong adult stock (a status reached for sampling locations SL4, SL5, and SL2) or a long lifespan (registered for sampling locations SL2 and IL12). For instance, despite favourable recruitment conditions in the Mira channel, adult population densities do not always correlate with these high recruitment rates. This discrepancy can be attributed to factors such as predation, competition and sediment displacement. In fact, high densities of adult *C. edule* and other suspension feeders can reduce settlement by ingesting settling larvae and juveniles or by smothering them through sediment displacement during burrowing and feeding activities [89]. André and Rosenberg [90] found that adults inhaled 75% of larvae at densities of 380 adults.m^−2^, a number that is widely surpassed in sampling locations SL2, IL12, SL4, and ML15 in descending order, leading to larval ingestion. However, de Montaudouin and Bachelet [89] noted that despite adults reject inhaled juveniles and close their syphons, the rejected juveniles usually die.

A critical concern is that the maximum cockle mean shell length in Ria de Aveiro remains below 26 mm, barely above the legal minimum catch size of 25 mm [91]. With sexual maturity being reached at 18.6 mm [3], individuals have little time to grow before facing harvesting pressure. Overexploitation at or below legal thresholds could erode reproductive capacity and long-term sustainability. Climate-driven shifts in size and growth [92] may further reduce viability. In this case, rather than altering the minimum catch size, it is more important to strengthen the control of catches and commercialisation, including stricter monitoring of compliance, improved enforcement against undersized sales, and adaptive management of harvesting intensity. Such measures would better ensure that individuals can reproduce before removal, thereby supporting the long-term sustainability of cockle populations in Ria de Aveiro. Adverse changes in these abiotic conditions, coupled with an intense harvesting pressure, could exacerbate the decline in population size and health. It is then crucial to comply with the minimum catch size, enforce seasonal harvesting bans, and enhance habitat conservation efforts, such as the delimitation of sanctuary areas [93], to ensure the sustainable exploitation of cockles in Ria de Aveiro. Our results highlight that protection efforts should be spatially targeted. Recruitment hotspots in the Mira channel, where settlement densities exceed 2000 recruits.m^−2^, act as essential nursery areas that warrant stricter protection from disturbance. In parallel, sites such as SL4, SL5, SL2, and IL12, which sustain the highest adult biomass and longest cohort persistence, function as reproductive strongholds. Prioritising these areas for enhanced conservation through seasonal closures would safeguard both the replenishment and reproductive capacity of the cockle population, strengthening the long-term sustainability of the fishery.

Additionally, research and monitoring programmes should be intensified to gather comprehensive data on cockle population dynamics. Such data are essential for making informed decisions that balance the needs of the local fishing community with the conservation of the cockle population. Adopting a science-based management approach, possibly through the creation of a cockle cooperative in this region, will safeguard the future of cockles in Ria de Aveiro while supporting the livelihoods that depend on this valuable resource.

## 5. Conclusions

The distribution and density of cockles in Ria de Aveiro are profoundly affected by environmental factors such as salinity, temperature, sediment composition, and currents velocity. Salinity influences cockle activity and survival, with optimal conditions found near the sea, where salinity is higher. Temperature also plays a critical role, as extremes can impact cockles’ health and population dynamics. Sediment type affects cockle distribution and recruitment, with a preference for muddy sand to sandy-mud substrates. Despite high recruitment peaks, adult populations do not always correlate with these peaks, possibly due to predation and sediment displacement. The proximity of the maximum mean shell length to the legal catch size raises sustainability concerns. In this way, the implementation of effective management measures, including adjustments to catch sizes and habitat conservation, and adaptive management that considers the intricate interplay of environmental factors and human activities are crucial for sustainable cockle populations in Ria de Aveiro.

## Figures and Tables

**Figure 1 biology-14-01427-f001:**
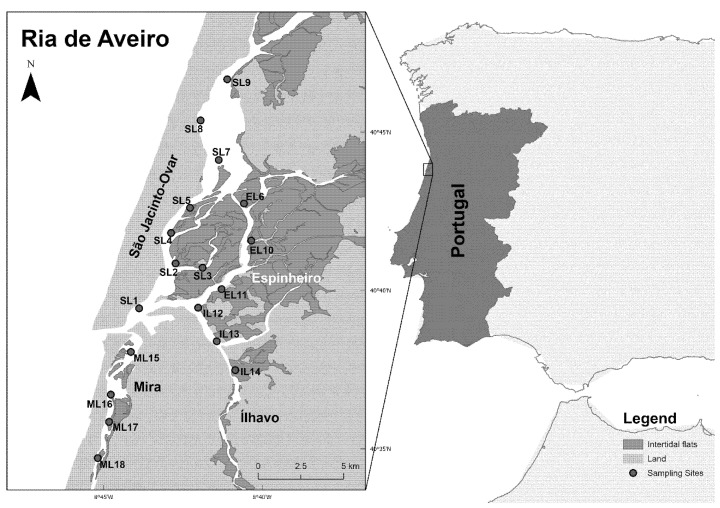
Geographic location of Ria de Aveiro, Portugal (40°37′20″ N; 8°44′22″ W) with reference to *Cerastoderma edule* sampling locations (blue spots, from L1 to L18) distributed along the four Ria de Aveiro’s channels: São Jacinto-Ovar (S), Espinheiro (E), Ílhavo (I), and Mira (M).

**Figure 2 biology-14-01427-f002:**
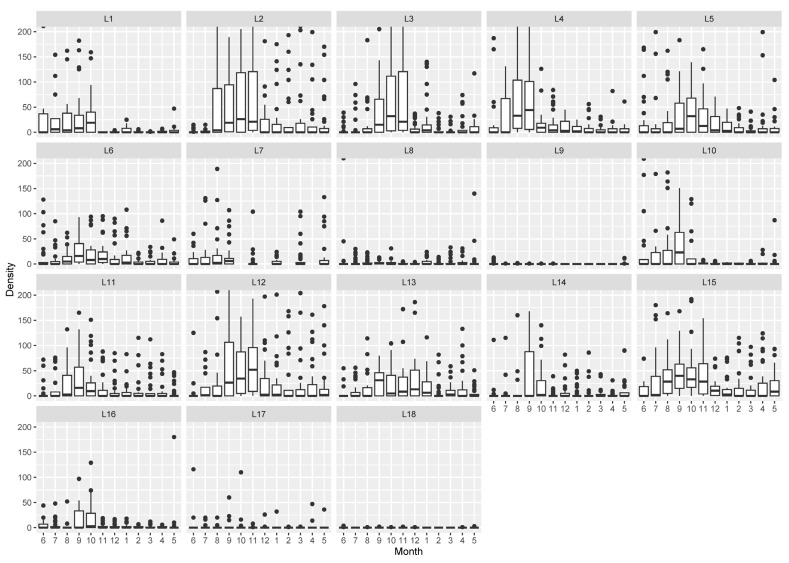
Cockle density per sampling location (from L1 to L18) with 12 months variation (from June 2020 [6] to May 2021 [5]).

**Figure 3 biology-14-01427-f003:**
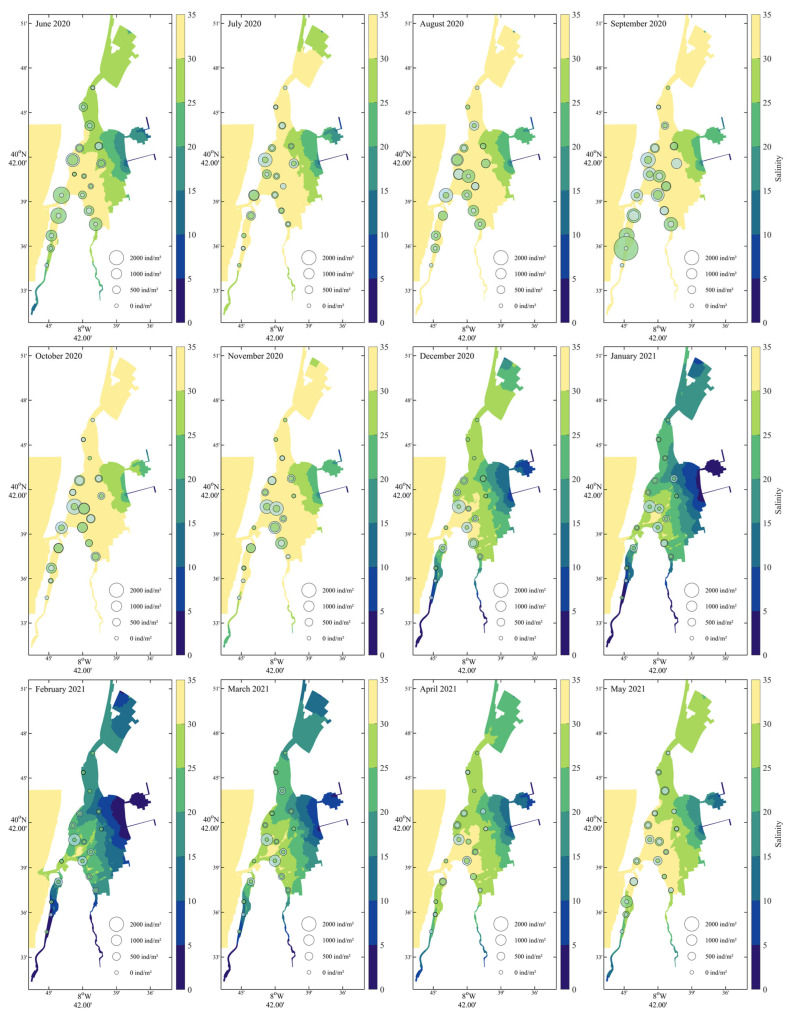
Monthly salinity distribution from June 2020 to May 2021 with *Cerastoderma edule* recruit and adult densities. Recruits (green bubbles) and adults (grey) are displayed with sizes scaled to density.

**Figure 4 biology-14-01427-f004:**
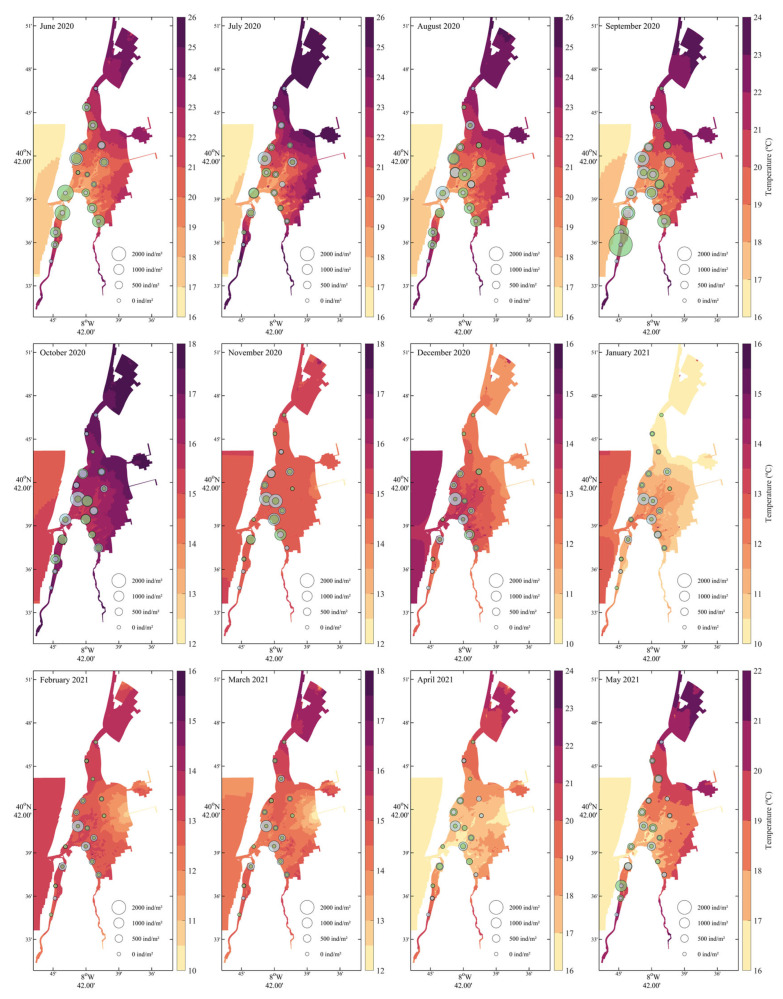
Monthly water temperature (°C) distribution from June 2020 to May 2021 with *Cerastoderma edule* recruit and adult densities. Recruits (green bubbles) and adults (grey) are displayed with sizes scaled to density.

**Figure 5 biology-14-01427-f005:**
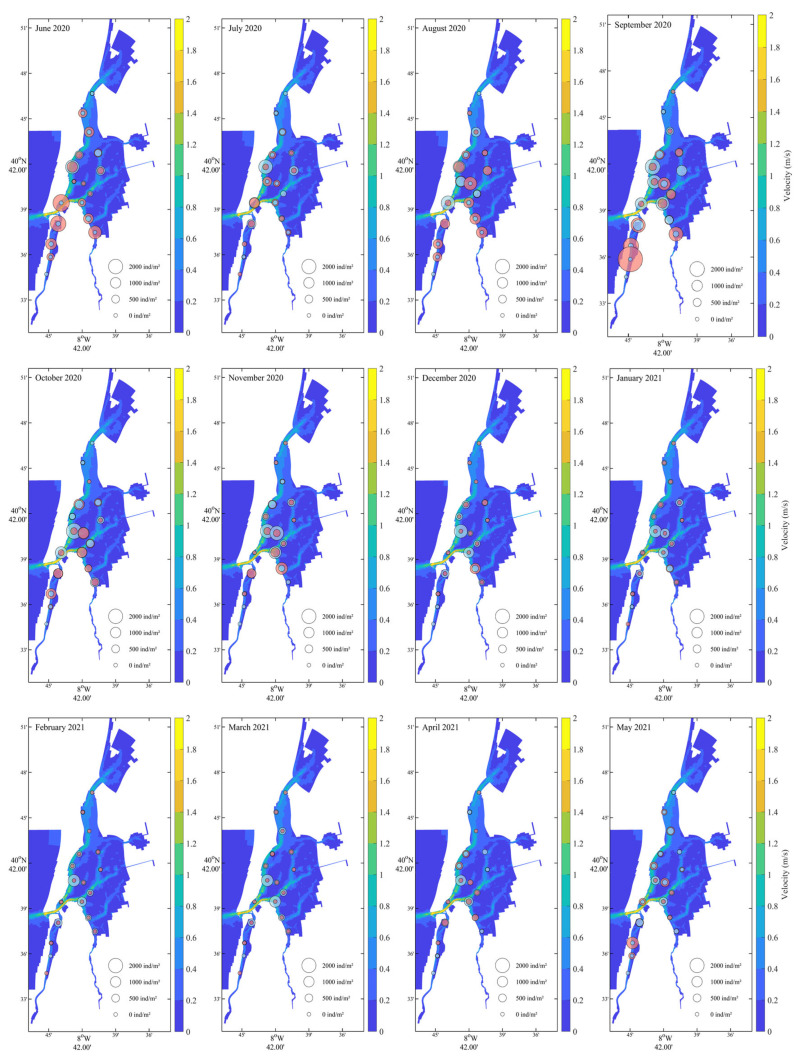
Monthly variation in hydrodynamic velocity (m/s) from June 2020 to May 2021 with *Cerastoderma edule* recruit and adult densities. Recruits (pink bubbles) and adults (grey) are displayed with sizes scaled to density.

**Figure 6 biology-14-01427-f006:**
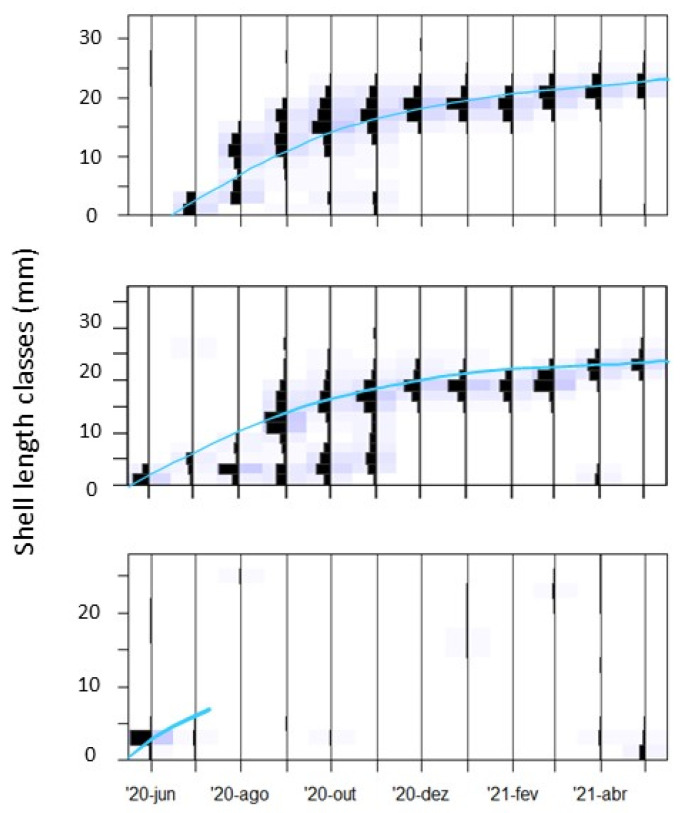
Length frequency histograms of sampling locations SL2 (top) and IL12 (middle), where cohorts presented the longest lifespans, and sampling location SL8 (bottom), as an example of an unsuccessful cohort. Grey bars represent confidence bands associated with the estimated parameters. Cohort duration is highlighted in light blue.

**Table 1 biology-14-01427-t001:** Mean values (±standard deviation [sd]) per sampling location (from L1 to L18, across four channels São Jacinto-Ovar—S, Espinheiro—E, Ílhavo—I, and Mira—M) for the monthly computed parameters: water salinity, water temperature (°C), current velocity (m/s), chlorophyll-a (µg/L), and submersion time (h).

Sampling Location	Salinity			Temperature			Current Velocity			Chlorophyll-a			Submersion Time		
	Mean	±	sd	Mean	±	sd	Mean	±	sd	Mean	±	sd	Mean	±	sd
SL1	31.5	±	2.8	16.1	±	2.5	0.26	±	0.00	1.5	±	1.2	730.0	±	21.6
SL2	30.6	±	3.4	16.3	±	2.9	0.43	±	0.01	1.6	±	1.3	730.0	±	21.6
SL3	30.8	±	3.4	16.5	±	3.1	0.18	±	0.01	1.5	±	1.3	469.6	±	13.4
SL4	30.0	±	3.8	16.5	±	3.1	0.72	±	0.01	1.6	±	1.3	730.0	±	21.6
SL5	29.0	±	4.4	16.8	±	3.5	0.68	±	0.01	1.6	±	1.4	730.0	±	21.6
EL6	24.8	±	6.2	17.1	±	4.1	0.16	±	0.00	2.1	±	1.7	730.0	±	21.6
SL7	28.2	±	4.9	17.3	±	4.0	0.50	±	0.01	1.7	±	1.5	730.0	±	21.6
SL8	27.7	±	5.2	17.7	±	4.5	0.45	±	0.00	1.7	±	1.5	730.0	±	21.6
SL9	27.0	±	5.6	18.1	±	5.0	0.86	±	0.01	1.8	±	1.6	730.0	±	21.6
EL10	27.2	±	5.1	16.6	±	3.4	0.45	±	0.01	1.9	±	1.5	730.0	±	21.6
EL11	30.1	±	3.7	16.1	±	2.7	0.97	±	0.01	1.7	±	1.3	730.0	±	21.6
IL12	31.2	±	3.1	16.0	±	2.5	0.61	±	0.01	1.5	±	1.2	730.0	±	21.6
IL13	30.7	±	3.5	16.3	±	2.9	0.49	±	0.01	1.5	±	1.3	730.0	±	21.6
IL14	28.6	±	5.4	16.8	±	3.5	0.25	±	0.00	1.8	±	1.3	730.0	±	21.6
ML15	30.5	±	4.5	16.3	±	2.8	0.09	±	0.00	1.7	±	1.2	557.2	±	15.9
ML16	25.0	±	9.7	17.4	±	4.1	0.39	±	0.00	3.0	±	2.6	730.0	±	21.6
ML17	22.5	±	11.4	17.9	±	4.6	0.47	±	0.00	3.9	±	3.5	730.0	±	21.6
ML18	20.0	±	12.7	18.2	±	4.9	0.47	±	0.00	5.3	±	4.8	730.0	±	21.6

**Table 2 biology-14-01427-t002:** Mean values (±standard deviation [sd]) per sampling location (from L1 to L18, across four channels São Jacinto-Ovar—S, Espinheiro—E, Ílhavo—I, and Mira—M) for the sediment descriptors evaluated: median grain size (MGS; ɸ) and total organic matter content (TOM; %) with respective classification systems.

Sampling Location	MGS				TOM			
	Mean	±	sd	Classification	Mean	±	sd	Classification
SL1	2.6	±	0.1	fine sand	2.2	±	0.6	medium
SL2	2.5	±	0.1	fine sand	1.9	±	0.7	low
SL3	1.9	±	0.2	medium sand	1.1	±	0.2	low
SL4	2.5	±	0.1	fine sand	2.6	±	1.3	medium
SL5	2.9	±	0.4	fine sand	3.4	±	1.4	medium
EL6	2.7	±	0.1	fine sand	2.3	±	0.5	medium
SL7	1.9	±	0.2	medium sand	1.9	±	0.5	low
SL8	1.5	±	0.2	medium sand	1.1	±	0.6	low
SL9	2.5	±	0.0	fine sand	1.0	±	0.5	low
EL10	2.6	±	0.2	fine sand	1.9	±	0.8	low
EL11	3.2	±	0.2	very fine sand	3.4	±	0.7	medium
IL12	2.3	±	0.1	fine sand	1.2	±	0.4	low
IL13	1.8	±	0.1	medium sand	1.2	±	0.6	low
IL14	3.0	±	0.3	fine sand	2.8	±	0.6	medium
ML15	1.5	±	0.1	medium sand	1.0	±	0.4	low
ML16	1.5	±	0.0	medium sand	0.9	±	0.5	very low
ML17	2.8	±	0.5	fine sand	3.0	±	1.0	medium
ML18	1.7	±	0.2	medium sand	1.5	±	0.6	low

**Table 3 biology-14-01427-t003:** Summary of fixed effects from Generalised Linear Mixed Models (GLMMs) predicting log-transformed cockle density (Total, Recruit, and Adult) in relation to environmental variables.

Predictor	Estimate	Std. Error	*p*-Value	Life Stage
Salinity	0.068	0.017	<0.001	Total
Temperature	0.072	0.028	0.013	Total
Salinity	0.070	0.032	0.032	Recruit
Temperature	0.280	0.046	<0.001	Recruit
MGS	0.742	0.405	0.073	Recruit
Current velocity	–2.384	1.200	0.063	Recruit
Month	0.154	0.044	<0.001	Recruit
Salinity	0.061	0.019	0.002	Adult
Temperature	–0.102	0.032	0.002	Adult
TOM	0.343	0.116	0.003	Adult

**Table 4 biology-14-01427-t004:** ELEFAN method results for cockle populations sampled in 18 locations of Ria de Aveiro (L1 to L18, across the four channels—São Jacinto-Ovar [S], Espinheiro [E], Ílhavo [I], and Mira [M]): von Bertalanfy growth function parameters L_∞_ (mm), K (year^−1^), and respective growth performance index (ɸ’). L—sampling location; NA—not available.

L	L_∞_	K	ɸ’
SL1	24.6	0.5	2.5
SL2	30.3	0.6	2.8
SL3	28.1	0.7	2.7
SL4	30.0	0.6	2.7
SL5	24.9	0.5	2.5
EL6	24.3	0.6	2.5
SL7	NA	NA	NA
SL8	26.3	0.5	2.6
SL9	NA	NA	NA
EL10	26.4	0.6	2.6
EL11	28.6	0.6	2.7
IL12	29.4	0.6	2.7
IL13	31.1	0.6	2.7
IL14	47.9	0.6	3.1
ML15	26.8	0.5	2.6
ML16	30.3	0.5	2.7
ML17	NA	NA	NA
ML18	NA	NA	NA

**Table 5 biology-14-01427-t005:** Mortality, stock size and stock biomass (g DW m^−2^) per sampling location, from L1 to L18, distributed along the 4 main channels: São Jacinto-Ovar (S), Espinheiro (E), Ílhavo (I), and Mira (M). NA—not available.

L	Mortality	Stock Size	Stock Biomass
SL1	1.10	23.06	6.79
SL2	1.03	79.25	18.82
SL3	1.15	21.77	5.64
SL4	1.84	288.12	106.63
SL5	1.21	116.64	35.32
EL6	1.17	66.54	19.32
SL7	NA	NA	NA
SL8	1.32	14.12	4.33
SL9	NA	NA	NA
EL10	1.10	63.33	15.96
EL11	0.94	41.51	9.48
IL12	0.92	61.06	13.66
IL13	1.10	22.42	6.24
IL14	1.07	2.73	1.11
ML15	0.93	79.18	18.81
ML16	1.20	5.37	2.19
ML17	1.01	0.59	0.85
ML18	NA	NA	NA

## Data Availability

Dataset available on request from the authors.

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
