# Peer review of "Cockle Population Dynamics in a Complex Ecological Aquatic System"

_biology, 2025, doi:10.3390/biology14101427_

Round 1
Reviewer 1 Report
Comments and Suggestions for Authors
Review report for paper COCKLE POPULATION DYNAMICS IN A COMPLEX ECOLOGICAL AQUATIC SYSTEM
The authors studied the population dynamics of Cerastoderma edule, the European edible cockle, within the coastal ecosystem of Ria de Aveiro, Portugal. This region is not only a hub of biodiversity but also crucial for cockle harvesting. The authors' study showed that environmental factors (water salinity, temperature, sediment characteristics, and current velocity) play a pivotal role in dictating the distribution and abundance of cockle populations. Their findings also brought to light a concern: the average maximum shell length of the cockles is approaching the legal minimum catch size, suggesting potential overexploitation. The results of this study may be important for the development of adaptive management strategies to ensure the sustainable exploitation of cockles in Ria de Aveiro. This includes considering alterations to catch sizes, implementing seasonal harvesting restrictions, and prioritizing habitat preservation.
I think this manuscript is well designed and well written and after a revision it can be published in the journal BIOLOGY.
I have made some comments which might improve the text.
- SIMPLE SUMMARY. I did not find this section in the text. The authors should provide a summary of their main findings in simple terms.
- In INTRODUCTION, the authors should provide information on the total catch of this species in Portugal along with its dynamics over the past decades.
- In INTRODUCTION, the authors should provide concrete examples of how C. edule contributes to biogeochemistry and biogeomorphology. What role does this species play in local trophic webs?
- In INTRODUCTION, the authors should specify which diseases are responsible for the decrease in the natural population of the European edible cockle.
- In INTRODUCTION, the authors should provide information on previous insights regarding cockle population dynamics and distribution. They should also explain how consistent monitoring of cockle populations over the long term can provide novel information. What does consistent long-term monitoring of cockle populations entail? We need to know if the authors' study meets the requirements for long-term monitoring.
- In MATERIAL AND METHODS, the authors should clarify whether their data met the assumptions for parametric two-way ANOVA such as normal data distribution and homogeneity of variance. Did the authors transform the data prior to this analysis?
- IN RESULTS, the authors mention some spatial trends in environmental parameters. Were these trends statistically significant? Could the authors provide p-vlues?
- IN RESULTS, the authors should check the SE value for salinity (Life stage: total). This value is larger than the estimate while for significant factors, standard errors must be lower than the Estimates.
- IN DISCUSSION, the authors should provide a clearer explanation of the negative role of temperature in shaping adult cockle density. What metabolic stress is assumed?
- IN DISCUSSION, the authors should compare the growth parameters of edible cockles in the study area with those in other regions during the discussion. They should also explain any differences they find.
- IN DISCUSSION, the authors should report the current catch size of edible cockles in the study area and provide recommendations on how to adjust this size to ensure the sustainability of cockle populations in Ria de Aveiro.
Author Response
REVIEWER #1
The authors studied the population dynamics of Cerastoderma edule, the European edible cockle, within the coastal ecosystem of Ria de Aveiro, Portugal. This region is not only a hub of biodiversity but also crucial for cockle harvesting. The authors' study showed that environmental factors (water salinity, temperature, sediment characteristics, and current velocity) play a pivotal role in dictating the distribution and abundance of cockle populations. Their findings also brought to light a concern: the average maximum shell length of the cockles is approaching the legal minimum catch size, suggesting potential overexploitation. The results of this study may be important for the development of adaptive management strategies to ensure the sustainable exploitation of cockles in Ria de Aveiro. This includes considering alterations to catch sizes, implementing seasonal harvesting restrictions, and prioritizing habitat preservation.
I think this manuscript is well designed and well written and after a revision it can be published in the journal BIOLOGY.
I have made some comments which might improve the text.
- SIMPLE SUMMARY. I did not find this section in the text. The authors should provide a summary of their main findings in simple terms.
Authors: The reviewer is right. We did not provide a simple summary in the first version of the manuscript. Please find it now in the revised version highlighted in yellow.
- In INTRODUCTION, the authors should provide information on the total catch of this species in Portugal along with its dynamics over the past decades.
Authors: The reviewer is right. This information is now available (L63-66).
- In INTRODUCTION, the authors should provide concrete examples of how C. edule contributes to biogeochemistry and biogeomorphology. What role does this species play in local trophic webs?
Authors: The information requested is in the new version of the manuscript (L74-80).
- In INTRODUCTION, the authors should specify which diseases are responsible for the decrease in the natural population of the European edible cockle.
Authors: Two examples were added in the text (L87-88).
- In INTRODUCTION, the authors should provide information on previous insights regarding cockle population dynamics and distribution. They should also explain how consistent monitoring of cockle populations over the long term can provide novel information. What does consistent long-term monitoring of cockle populations entail? We need to know if the authors' study meets the requirements for long-term monitoring.
Authors: The information requested is in the new version of the manuscript (L107-116).
- In MATERIAL AND METHODS, the authors should clarify whether their data met the assumptions for parametric two-way ANOVA such as normal data distribution and homogeneity of variance. Did the authors transform the data prior to this analysis?
Authors: The information requested is in the new version of the manuscript (L204-205).
- IN RESULTS, the authors mention some spatial trends in environmental parameters. Were these trends statistically significant? Could the authors provide p-vlues?
Authors: The description of environmental variables in section 3.1 is merely descriptive. It was not statistically tested for lack of statistical power, i.e. number of replicates. However, this trend is modelled and showed in the figures 3, 4 and 5.
- IN RESULTS, the authors should check the SE value for salinity (Life stage: total). This value is larger than the estimate while for significant factors, standard errors must be lower than the Estimates.
Authors: The reviewer is absolutely right, this was a transcription error. The corrected values are highlighted in yellow in Table 3.
- IN DISCUSSION, the authors should provide a clearer explanation of the negative role of temperature in shaping adult cockle density. What metabolic stress is assumed?
Authors: Our results reflect this thermal sensitivity, since adult cockle densities decline with higher temperatures, likely due to increased metabolic costs associated with elevated respiration rates, reduced energy available for growth and reproduction, and higher mortality under thermal stress. The statement was revised and improved with an example (L505-508).
- IN DISCUSSION, the authors should compare the growth parameters of edible cockles in the study area with those in other regions during the discussion. They should also explain any differences they find.
Authors: This comparison and the differences found are now included in the manuscript (L541-555).
- IN DISCUSSION, the authors should report the current catch size of edible cockles in the study area and provide recommendations on how to adjust this size to ensure the sustainability of cockle populations in Ria de Aveiro.
Authors: The minimum catch size was already included in the first manuscript version (L621) and now we added a recommendation (L625-630).
Reviewer 2 Report
Comments and Suggestions for Authors
The manuscript represents robust and very well-written research. All sections are very well presented. ABSTRACT may present a greater contribution of RESULTS. INTRODUCTION addresses biological and ecological aspects of the species. MATERIALS AND METHODS provides a good description of the study area and the analyses used. RESULTS describe in detail what was proposed. DISCUSSION is very good and describes all the relevant aspects.

Author Response
REVIEWER #2
L14-17 I believe the authors could describe more results
Authors: The reviewer is right, we added more results in this section (L32-39).
Keywords: Cardiidae
Authors: This word is now part of the keywords list (L11).
L55 Sousa et al., 2017
Authors: We believe the reviewer is suggesting that the references be listed in chronological order. However, the references are currently formatted according to the journal’s requirements, which follow a numbered style.
L69 Ricardo et al., 2017
Authors: We believe the reviewer is suggesting that the references be listed in chronological order. However, the references are currently formatted according to the journal’s requirements, which follow a numbered style.
L140 included
Authors: Corrected (L180).
L203 &
Authors: The references are currently formatted according to the journal’s requirements, which follow a numbered style.
L238 ,
Authors: Comma removed (L289).
L350 Temperature was significantly negatively correlated
Authors: We apologize, but we were unable to understand the reviewer’s intent. We have therefore left the sentence unchanged (L412).
L404 4.1 Salinity and Temperature Effects
Authors: The title was removed.
L408 Peteiro et al., 2018
Authors: We believe the reviewer is suggesting that the references be listed in chronological order. However, the references are currently formatted according to the journal’s requirements, which follow a numbered style.
L432 Ramón, 2003
Authors: We believe the reviewer is suggesting that the references be listed in chronological order. However, the references are currently formatted according to the journal’s requirements, which follow a numbered style.
L434 Parada et al., 2012; Parada and Molares, 2008
Authors: We believe the reviewer is suggesting that the references be listed in chronological order. However, the references are currently formatted according to the journal’s requirements, which follow a numbered style.
L453 and
Authors: The references are currently formatted according to the journal’s requirements, which follow a numbered style.
L459 Kater et al., 2006
Authors: We believe the reviewer is suggesting that the references be listed in chronological order. However, the references are currently formatted according to the journal’s requirements, which follow a numbered style.
L462 4.2 Population Structure
Authors: The title was removed.
Discussion: The authors could better discuss the "population structure" based on other studies
Authors: Thanks for your comment. Following this recommendation, authors added a comparison between our results and other studies (L541-555).
L484 4.3 Recruitment Patterns
Authors: The title was removed.
L550 4.4 Conservation Implications
Authors: The title was removed.
Reviewer 3 Report
Comments and Suggestions for Authors
Overall Summary:
The manuscript is well-written, easy to understand, and relevant. The research topic is considered actual and valuable, providing significant insights into the population structure and production characteristics of cockles in the region. The core findings - that environmental conditions like water salinity and temperature are primary drivers of cockle (Cerastoderma edule) density and the subsequent proposal of conservation zones — are important contributions.
However, to meet publication standards, several major revisions are required to enhance the clarity of the manuscript.
Abstract: The abstract must be expanded to include a clear statement of the research aim. Furthermore, it should present the main quantitative results, not just qualitative findings.
Introduction: The final paragraph (specifically lines 85-88) lacks a clear and concise statement of the research aim. The last paragraph of the introduction should include a direct sentence that outlines the specific objectives of the research, the key parameters investigated.
Methods: The methodology is currently incomplete in two critical areas. First, there is no list of the parameters used for the population analysis. Second, the description of the specific biological parameters measured from the collected cockle samples is entirely absent. Please, include a new subsection to the Methods to detail all parameters measured from the cockle samples.
The detailed procedures for the cohort analysis, growth modelling, and the calculation of stock size and biomass (which appear in Results sections 3.2.2 to 3.2.4) are not described in the Methods section.
Results: It is unclear if any predicted environmental parameters from models were verified against actual, measured field data. Evidence is required to confirm that the model provides adequate and accurate values.
On line 275, the life stages of the cockles are mentioned, but the method for determining these stages is not explained.
On line 284, the use of "month" as a categorical variable influencing density is questioned. The reviewer notes that a calendar month is a human construct and is likely a proxy for underlying environmental drivers like temperature or light cycles. The authors should clarify their use of "month" in the text and justify its use as a proxy for integrated seasonal changes or, alternatively, replace it with direct, measured environmental variables that more accurately represent the driving ecological factors.
Author Response
REVIEWER #3
The manuscript is well-written, easy to understand, and relevant. The research topic is considered actual and valuable, providing significant insights into the population structure and production characteristics of cockles in the region. The core findings - that environmental conditions like water salinity and temperature are primary drivers of cockle (Cerastoderma edule) density and the subsequent proposal of conservation zones — are important contributions.
However, to meet publication standards, several major revisions are required to enhance the clarity of the manuscript.
Abstract: The abstract must be expanded to include a clear statement of the research aim. Furthermore, it should present the main quantitative results, not just qualitative findings.
Authors: We agree with this suggestion, the abstract was completed and reformulated accordingly (L29-39).
Introduction: The final paragraph (specifically lines 85-88) lacks a clear and concise statement of the research aim. The last paragraph of the introduction should include a direct sentence that outlines the specific objectives of the research, the key parameters investigated.
Authors: The final paragraph was rewritten to clearly state the research aim and key parameters investigated (L117-122).
Methods: The methodology is currently incomplete in two critical areas. First, there is no list of the parameters used for the population analysis. Second, the description of the specific biological parameters measured from the collected cockle samples is entirely absent. Please, include a new subsection to the Methods to detail all parameters measured from the cockle samples.
Authors: Starting in order of occurrence, biological parameters evaluated were already described in the first manuscript version but are now evidenced in a separated paragraph (L155-160). Regarding the parameters used for the population analysis, they are now listed in the Methods section, subsection 2.3.2 (L228-230).
The detailed procedures for the cohort analysis, growth modelling, and the calculation of stock size and biomass (which appear in Results sections 3.2.2 to 3.2.4) are not described in the Methods section.
Authors: Cohort analysis, growth modelling, and the calculation of stock size and biomass are described in the Methods section, subsection 2.3.2.
Results: It is unclear if any predicted environmental parameters from models were verified against actual, measured field data. Evidence is required to confirm that the model provides adequate and accurate values.
Authors: We thank the reviewer for this valuable comment. We have clarified in Section 2.2 that the model applied was validated for Ria de Aveiro (Picado et al., 2024). The revised text now explicitly states the validation process and the variables considered (L170-174).
On line 275, the life stages of the cockles are mentioned, but the method for determining these stages is not explained.
Authors: The method was described in the first version of the manuscript, but we agree that it was not clearly explained how cohort separation was translated into life-stage assignments. This process is now clarified in the revised version (L236-240).
On line 284, the use of "month" as a categorical variable influencing density is questioned. The reviewer notes that a calendar month is a human construct and is likely a proxy for underlying environmental drivers like temperature or light cycles. The authors should clarify their use of "month" in the text and justify its use as a proxy for integrated seasonal changes or, alternatively, replace it with direct, measured environmental variables that more accurately represent the driving ecological factors.
Authors: We agree that “month” is a human-defined categorical variable that often correlates with environmental parameters, particularly temperature. While “month” was initially included in all models, it was typically excluded following VIF analysis. For recruit density, however, “month” neither showed multicollinearity nor a strong correlation with other predictors and was therefore retained. In this context, “month” likely captures seasonal effects on recruit density patterns that are not fully explained by the other variables. We chose to keep “month” in the model because it improved model fit.
Round 2
Reviewer 3 Report
Comments and Suggestions for Authors
The Authors provide thorough and thoughtful responses for all comments. The revision have significantly improved the quality and clarity of the manuscript. I have no questions